# Mechanical Stress Improves Fat Graft Survival by Promoting Adipose-Derived Stem Cells Proliferation

**DOI:** 10.3390/ijms231911839

**Published:** 2022-10-06

**Authors:** Jeong Jin Chun, Jiyeon Chang, Shindy Soedono, Jieun Oh, Yeong Jin Kim, Syeo Young Wee, Kae Won Cho, Chang Yong Choi

**Affiliations:** 1Department of Plastic and Reconstructive Surgery, Soonchunhyang University Hospital, Gumi 39371, Korea; 2Department of Integrated Biomedical Science, Soonchunhyang Institute of Medi-Bio Science (SIMS), Soonchunhyang University, Cheonan 31151, Korea; 3Department of Medical Biotechnology, Soonchunhyang University, Asan 31583, Korea; 4Department of Plastic and Reconstructive Surgery, Soonchunhyang University Hospital, Bucheon 14584, Korea

**Keywords:** adipose-derived stem cells, mechanical stress, cell-assisted lipotransfer, stem cell proliferation

## Abstract

Cell-assisted lipotransfer (CAL), defined as co-transplantation of aspirated fat with enrichment of adipose-derived stem cells (ASCs), is a novel technique for cosmetic and reconstructive surgery to overcome the low survival rate of traditional fat grafting. However, clinically approved techniques for increasing the potency of ASCs in CAL have not been developed yet. As a more clinically applicable method, we used mechanical stress to reinforce the potency of ASCs. Mechanical stress was applied to the inguinal fat pad by needling. Morphological and cellular changes in adipose tissues were examined by flow cytometric analysis 1, 3, 5, and 7 days after the procedure. The proliferation and adipogenesis potencies of ASCs were evaluated. CAL with ASCs treated with mechanical stress or sham control were performed, and engraftment was determined at 4 weeks post-operation. Flow cytometry analysis revealed that mechanical stress significantly increased the number as well as the frequency of ASC proliferation in fat. Proliferation assays and adipocyte-specific marker gene analysis revealed that mechanical stress promoted proliferation potential but did not affect the differentiation capacity of ASCs. Moreover, CAL with cells derived from mechanical stress-treated fat increased the engraftment. Our results indicate that mechanical stress may be a simple method for improving the efficacy of CAL by enhancing the proliferation potency of ASCs.

## 1. Introduction

Autologous fat grafting is a widely used procedure for soft tissue volume augmentation owing to its biocompatibility, natural texture, and versatility [1,2,3]. Despite these advantages, the engraftment rate of fat grafts ranges from 20–80% [3,4], which is the main limitation of fat grafting in reconstructive surgery. For example, patients with hemifacial microsomia and patients who want aesthetic augmentation of breast or buttock should have undergone massive and repeated fat grafting [5,6,7,8]. This is because of the lower efficacy of fat graft survival and the limited amount of fat. Therefore, it is important to develop procedures to resolve the unpredictable and low survival rate of fat grafts.

Numerous studies have been conducted to improve fat grafts survival rate. In terms of surgical procedures, there have been attempts to do the following: harvesting using syringe-aspiration, avoiding lidocaine use, performing centrifugation, and washing harvested grafts with nutrient-containing solutions; however, these modified procedures have poor reproducibility and results [9,10,11,12,13]. It has also been reported that the addition of vascular endothelial growth factors (VEGFs), platelet-derived growth factors, or platelet-rich plasma improves transplantation results [2,14,15]. Some researchers have shown that preconditioning by injecting drugs such as deferoxamine or tamoxifen increases fat graft retention following grafting [16,17]. However, the main limitation of these methods is that drug or growth factor injections must be repeated several times over several weeks, which raises concerns regarding their safety for use in humans. 

Recently, cell-assisted lipotransfer (CAL), a fat graft mixed with stromal vascular cells (SVC), including adipose-derived stem cells (ASCs), has been reported [14,15,16,17,18,19]. ASCs are one of mesenchymal stem cells (MSCs) that are able to differentiate into several lineages including adipocytes. ASCs also have potential to secrete various functional molecules such as cytokines, growth factors, and immunomodulators. Compared to other tissue-derived stem cells, ASCs have unique advantages of extensive tissue sources, easy availability, and lower donor site morbidity [20,21,22]. Based on the characteristics of ASCs, addition of ASCs are considered to be the beneficial effects on the tissue repair and homeostasis. Indeed, several randomized controlled trials and meta-analyses have demonstrated that the graft retention of ASC-enriched fat grafts is superior to that of conventional fat grafts [3,23,24,25], indicating that CAL is an intervention technique to increase fat graft survival. However, the current CAL has some limitations in terms of clinical application and achievement of maximal efficacy. For example, a significantly higher amount of adipose tissue is required for CAL, which would be a practical issue for slim patients with limited adipose tissue in donor sites. It might also be necessary to isolate ASCs and culture them in vitro to achieve a sufficient number of cells, which raises safety and cost concerns. Another concern is the heterogeneity in ASCs which would generate the wide range of stem cell quality including stemness, proliferation, and differentiation potentials [22]. Difficulty in the preservation of specific ASC phenotype during the procedure is another important issue [26]. Thus, it is necessary to find a way to improve the ASC potential to overcome the limited number of ASCs in adipose tissue. 

Mechanical stimulation has been implicated as a stem cell potential regulator. Several studies have indicated that mechanical force plays an important role in regulating cell growth and proliferation, and an appropriate mechanical stimulation treatment could promote the proliferative capacity of bone marrow stem cells [27,28,29,30]. Ouyang et al. reported that mechanical stimulation promotes the proliferation of mesenchymal stem cells (MSCs) [31]. On the contrary, several studies have shown that mechanical stimulation has no effect on cell proliferation, nor does it reduce mesenchymal stem cell proliferation [32,33,34,35]. These controversial results could be ascribed to the diversity of experimental conditions, including the specific mechanical stimulation, stem cell culture conditions, and the wide range of loading parameters used. Importantly, the effect of mechanical stimulation on the proliferative and differentiation capacities of ASCs in vivo has not been previously studied. 

As a more clinically applicable method, in the current study, we incorporated mechanical stress into donor fat in a murine model, with needling prior to fat harvesting. Next, we explored the role of mechanical stress on the ASC potential in vivo and in vitro. This study also investigated whether an in vivo model of CAL conducted with SVC after mechanical stress showed better engraftment than conventional CAL. 

## 2. Results

### 2.1. Effects of Mechanical Stress on Adipose Tissue Histology and Cellular Components

To investigate the effects of mechanical stress on donor adipose tissues, we performed mechanical stress or sham procedures on the right and left sides of subcutaneous fat in the same mice and then harvested the adipose tissues at different time points (Figure 1A). As expected, mechanical stress induced the infiltration of inflammatory cells around adipocytes on day 1 and elevated immune cells in adipose tissue until day 7. However, perilipin staining demonstrated that mechanical stress did not affect the number of viable adipocytes until day 5, despite a marginal reduction at day 7. In addition, vessel staining showed that vessel numbers around the adipocytes were slightly increased on days 3 and 5 post-mechanical stress (Figure 1B). These histological assessments demonstrated that mechanical stress-treated donor adipose tissue still contains relevant numbers of viable adipocytes and vessels, which are important factors in regulating fat graft survival.

Since adipose tissues contain a wide variety of cell types, including adipocytes, endothelial cells (ECs), and adipose-derived stem cells (ASCs), which can affect adipose tissue remodeling and regeneration, we next explored the cellular components contributing to these effects by flow cytometry analysis. 

As depicted in Figure 1C, the cellular population of leukocytes, ECs, and ASCs in stromal vascular cells (SVC) from adipose tissues was determined using a combination of several markers including CD45 (leukocyte marker), CD31 (endothelial cell marker), Sca1, and CD140a (stem cell markers), leukocytes (CD45^+^), ECs (CD45^−^CD31^+^), and ASCs (CD45^−^CD31^−^Sca-1^+^CD140a^+^). In line with the histological results, mechanical stress markedly increased leukocyte frequency in adipose tissue on day 1 and was sustained until day 7 (Figure 1D). Mechanical stress also enhanced ECs frequency by ~2 folds at day 1 and gradually, but not significantly, increased ECs frequency until day 7 (Figure 1E). Moreover, the frequency of ASCs was immediately induced at day 1 and showed significant enrichment of ASCs on day 5 post-treatment (Figure 1C), indicating that mechanical stress could be a novel way to increase ASCs concentration in adipose tissue in vivo.

To examine whether mechanical stress could affect the in vivo ASC proliferation potential, we further analyzed proliferating cell populations using a Ki-67 proliferation marker. In the control fat (day 0), most ASCs were quiescent and less than 2% were Ki67^+^ proliferating cells. However, in the mechanical stress-treated fat, proliferating ASCs populations were markedly upregulated at day 3, reached statistical significance at day 5, and were then maintained until day 7 (Figure 2B). Collectively, these results demonstrate that mechanical stress promotes ASCs proliferation, which leads to the enrichment of ASCs in fat, especially on day 5. 

### 2.2. Effects of Mechanical Stress on Adipose-Derived Stem Cell Proliferation and Differentiation Potentials

Enrichment of ASC numbers and increase in the proliferative capacity of ASCs in mechanical stress-induced fat in vivo led us to investigate whether mechanical stress could directly regulate ASC cell potency. Using isolated ASCs treated with mechanical stress or sham, proliferation and adipocyte differentiation potential were examined. The 3-(4,5-Dimethylthiazol-2-yl)-2,5-diphenytetrazolium bromide (MTT) assay demonstrated that the proliferation of ASCs isolated from mechanical stress-treated adipose tissue was significantly higher than that of control ASCs at days 3 and 4, while cytotoxicity was similar between groups, as indicated by equivalent MTT activity between groups at day 0 (Figure 3A). Assessment of viable cell number counting also showed a higher proliferating potential of ASC isolated from mechanical stress-treated adipose tissue (Figure 3B). Altogether, mechanical stress imprints a higher proliferative capacity in ASCs, which is still sustained in the absence of stress or in vitro conditions.

The adipocyte differentiation potential of ASC was determined by the accumulation of lipid droplets in adipocytes and adipocyte-specific gene expression (Figure 3C,D). The differentiated adipocyte levels assessed by Oil Red O staining demonstrated that the differentiation level of ASCs treated with mechanical stress was comparable to that of the control (Figure 3C). Gene expression of peroxisome proliferator-activated receptor-γ (*Pparg*), CCAAT/enhancer-binding protein-α (*Cebpa*), fatty acid-binding protein 4 (*Fabp4*), and fatty acid synthase (*Fasn*) were similar between the groups (Figure 3D). Supporting these results, immunoblotting against specific antibodies for adipogenesis marker showed that protein expressions of PPARγ, C/EBPα, and FABP4 between mechanical stress and control group during adipocyte differentiation were comparable. Overall, these results indicate that mechanical stress-treated ASCs have similar preserved differentiation potential as ASCs. 

### 2.3. Effects of Mechanical Stress-Induced Adipose-Derived Stem Cells on Fat Graft Survival

Based on the observation of increased ASC proliferation potential in vitro and in vivo, we examined whether mechanical stress-induced alteration of ASCs potency in fat would affect the fat graft survival rate. At day 5 post-mechanical stress treatment, SVC were isolated from inguinal fat, mixed with sham fat, and then transplanted into mice (Figure 4A). At 4 weeks post-transplantation, the fat retention weight and volume were not significantly different (data not shown). However, H&E staining showed that the D5 SVC-mixed fat graft had more adipocytes and less fibrosis than the control (Figure 4B). Furthermore, perilipin staining demonstrated that the D5 SVC-mixed fat graft contained a higher number of viable adipocytes (Figure 4C). These CAL results indicate that preconditioning of donor fat with mechanical stress improves the fat graft survival of CAL and imply that mechanical stress could be a novel method for increasing the fat graft survival rate of CAL.

### 2.4. Discussion

CAL is a new cosmetic and reconstructive procedure for fat grafting, and multiple approaches have been used to increase its efficacy. While supplemented ASC numbers are very well known as the major regulatory factor in determining the high survival rate of fat grafts [25], the optimal concentration of ASCs in fat grafts has not been established [36,37,38], mainly due to the wide range of ASC cell quality in cell-assisted transfer. Moreover, in clinical applications, ASC numbers from donor fat tissue are limited; thus, additional modifications, such as excess fat tissue harvesting and in vitro ASC culture, are needed [39,40]. The development of a method to increase ASC potential could solve the problem of the current CAL procedure. In this study, we devised a new method using mechanical stress to reinforce the potency of ASCs to increase the efficacy of ASC-enriched fat grafts. The current study demonstrates that mechanical stress on donor fat improves ASC-enriched fat graft survival by increasing the proliferating capacity of ASCs. These results indicate that applying mechanical stress to donor fat is a novel technique for resolving the limitations of current cell-assisted transfer.

In the stem cell-based therapy such as CAL, it is important to create an environment that maintains the differentiation and proliferation capacity of ASCs [41]. After the discovery showing that the differentiation of MSC is directed by the hydrogel elasticity level, biophysical cues are recognized as a critical factor to regulate stem cell behaviors and fate [42]. Among the various factors in the biophysical cues, the modulatory role of mechanical stress on the potency of ASCs has been addressed in an in vitro model [29,43,44,45]. Mechanical activation and mechanical stretching have been shown to potentiate the proliferative ability of stem cells [46]. In addition, MSCs and fibroblasts have been identified as appropriate mechanical stimulatory parameters for regulating proliferation [28]. In the aspect of differentiation potential, accumulating evidence have shown that the mechanical microenvironments including stiffness and tension influence the osteogenic- and adipogenic-differentiation of ASCs [47,48,49]. In addition to the level of mechanical stress, rate of stress relaxation is also identified as the regulator of differentiation of MSCs [50]. Taken together, these findings that external mechanical factors affect the proliferation and differentiation of MSCs in vitro infers that mechanical stimulation can enhance the potency of ASCs in vivo [47]. However, there are few studies demonstrating the effects of mechanical cues on the ASC potential and fate in vivo. In this study, we revealed mechanical stress significantly increased the proliferation of ASCs in fat (Figure 1 and Figure 2), supporting the in vivo function of mechanical activation on stem cell ability. To the best of our knowledge, this is the first study to show an increase in the proliferation potential of ASCs by mechanical stress. A recent in vitro study demonstrated that a memory of past mechanical cues in cells can regulate behavior of the stem cells over time [51,52]. Consistently, we observed that the increased proliferation potential of ASCs was induced by mechanical stimulation and sustained in vitro, indicating that mechanical stress directly regulates ASC proliferation potential and the existence of imprinted memory in ASC by mechanical stimulation (Figure 3). Importantly, through the in vivo CAL model, the improvement of ASCs potency by mechanical stress was valid even when applied to CAL.

Besides MSC cell behavior itself, dynamic regulation of ASCs with niche is crucial factor to determine the fat graft survival rate. Previously, our group reported that VEGF-induced angiogenesis and lipolysis play an important role in regulating fat graft survival [53,54]. Recent studies have shown that biophysical signals can regulate the immunomodulatory functions of ASCs through cytokine secretion [55]. It is also reported that stiffness of the extracellular matrix controls adipocyte differentiation and lipolysis [56]. Collectively, these results suggest that improvement of fat graft enriched with ASCs from mechanical stimulation could be in part attributed to the changes in the interactions of ASCs with surrounding niches. In addition, the performance of implanted stem cells depends not only on proliferation and differentiation but also on migration, adhesion, and paracrine secretion, which deserve to be explored in depth in the future. 

Intriguingly, the characteristics of the improvement of fat grafts by mechanical stress are similar to the enhancement of regenerative ability by the surgical delay procedure. The surgical delay procedure enhances flap viability by promoting regeneration properties, including progenitor cell participation [57]. Delayed flaps have been reported to increase the proliferation of mesenchymal stem cells [37]. Suga et al. also found that the key mechanism of the reinforced function of ASCs after ischemia is an increase in angiogenic growth factors such as fibroblast growth factor 2, hepatocyte growth factor, and platelet-derived growth factor [58,59]. Similarly, mechanical stress enhances the proliferation of ASCs, angiogenesis, and endothelial cell numbers in fat (Figure 1). Whether the molecular mechanisms underlying the promotion of stem cell proliferation by mechanical stress would be the same as those by flap delay or the involvement of angiogenesis through endothelial cells in these procedures warrants further study. 

The current study has been highly relevant to unmet clinical needs. First, mechanical stress does not require additional steps to increase ASCs in fat; thus, it can resolve the problems associated with current CAL procedures. In particular, it is very important to increase the potency of ASCs because the patient needs repeated massive fat grafting to create facial symmetry in case of hemifacial microsomia [5,6,60]. Second, the mechanical stress does not comply with a safety issue, whereas supplementation with recombinant growth factor(s) causes clinical concerns regarding tumors. Third, in vivo validation of the improvement of CAL efficacy by targeting ASC behavior can bridge the stem cell-based translational research gap. A lot of efforts to regulate stem cells, particularly by various biomaterials and their nano-architected structures, have been made. Recent studies have shown that advances in the biomaterial- and nano-technologies can provide the temporal and narrow-range of mechanical signal control to regulate ASC fate and behavior [42,48,61,62], which make it possible to establish more appropriate procedure to apply to the clinic investigation. Thus, the application of specific physical factors can propel the specific phenotype of ASCs potentials in more clinically favorable and specialized direction.

Despite these findings regarding a novel technique to improve fat grafting, this study has several limitations. First, needling as a mechanical stimulant may be too traumatic for the subjects. However, the microenvironment of fat tissue after needling showed that needling did not cause permanent damage. However, the administration of needling in human patients can cause ethical issues. Therefore, further studies are needed to find another mechanical stimulant that can replace needling and show the same results as ours. Second, the molecular mechanisms underlying the induction of proliferation potential by mechanical stimulation have not been completely elucidated in this study. It has been reported that PI3K/AKT and MAPK signaling pathway participate in the effects of mechanical stretch on the biological characteristics of human ASCs [46]. Focal adhesion signaling has been implicated as a regulator of mechanical signaling and proliferation [63,64]. Future studies regarding molecular regulators and relevant biomolecules in mechanical stimulation-induced proliferation of stem cells are warranted. 

In summary, we demonstrated that donor site mechanical stress not only increases the absolute number of ASCs, but also enhances the proliferation of ASCs in adipose tissue without affecting the differential potential of ASCs. Our findings on the improvement of fat graft survival using mechanical stress-treated adipose tissue in both CAL and whole fat grafts indicate that donor site mechanical stress could enhance fat graft survival by increasing the potency of ASCs in fat. These results suggest that mechanical stress could be a novel technique to improve the efficacy of ASC-enriched fat grafts.

## 3. Materials and Methods

### 3.1. Animal Studies

C57BL/6 mice were purchased from Orient Bio (Seongnam, Korea) and were housed at 23 °C on a 07:00–19:00 light cycle in specific pathogen-free animal facilities at Soonchunhyang Institute of Medi-bio Science (SIMS). All experiments and surgical preparations were performed according to a protocol approved by the Soonchunhyang University Animal Care and Use Committee (IACUC number: SCH20-0074, date of approval date: 11 December 2020).

Mechanical stress was induced using a 21-gauge needle, with 25 needling on the right unilateral inguinal fat pad and the contralateral inguinal fat pads used as the control (Figure 1A). After mechanical stress, the microenvironment of the donor site fat tissue was analyzed in a time-dependent manner on days 0, 1, 3, 5, and 7 with histological staining and flow cytometry analysis (*n* = 6~8 for each group).

For the cell-assisted lipotransfer (CAL), normal chow-fed C57BL/6 mice at 8 weeks of age were used as recipients. For fat transplantation, 300 mg dissected fat from C57BL/6 mice was mixed with 1 × 10^6^ stromal vascular cells (SVC) from the fat of mice pretreated with surgical preconditioning procedures or from the control mice (Figure 4A). The mixture was then injected into the backs of C57BL/6 mice via a 5-mm incision and sutured using 4–0 silk (*n* = 4 per group). Four weeks after fat transplantation, the mice were euthanized, and the grafts were harvested and subsequently analyzed, as described later. No mortality was observed in the animals following fat transplantation in the present study.

### 3.2. Histological Analysis

Adipose tissues and fat grafts were fixed in 4% paraformaldehyde, processed, and embedded in paraffin. The samples were then sectioned at a thickness of 5 µm and stained with hematoxylin and eosin (H & E). For immunohistological analysis, tissue sections were incubated with goat anti-perilipin (Abcam, Boston, MA, USA) and isolectin B4 (Vector Laboratories, Newwork, CA, USA), followed by incubation with a horseradish peroxidase-conjugated secondary antibody (Vector Laboratories, Burlingame, CA, USA) or Alexa Fluor 568-conjuaged secondary antibodies (Invitrogen, Waltham, MA, USA). For Oil Red O staining, cells were fixed with 4% paraformaldehyde, rinsed with deionized water, and incubated with Oil Red O staining solution (5 ng/mL) for 5 min. Images were captured using a Leica DM1000 light microscope (Leica Microsystems, Wetzlar, Germany) or a Leica DMi8 fluorescence microscope (Leica Microsystems, Wetzlar, Germany). 

### 3.3. Isolation of Stromal Vascular Cell and Flow Cytometry Analysis

Stromal Vascular cells (SVC) from the donor site after mechanical stress were isolated as previously described [65]. Briefly, adipose tissues were digested in phosphate-buffered saline (PBS) with 0.5% BSA and 1 mg/mL type II collagenase for 25 min at 37 °C, and stromal vascular cells (SVC) were separated from adipocytes by centrifugation. Isolated SVC were stained using live/dead fixable dyes (Invitrogen): anti-CD45 (eBioscience, 30-F11), anti-Ly-6A/E(Sca1) (BioLegend, D7, San Diego, CA, USA), anti-CD31 (BioLegend, 390, San Diego, CA, USA), and anti-CD140a (BioLegend, APA5, San Diego, CA, USA) and anti-Ki67 (eBioscience, SolA15, San Diego, CA, USA). Cells were analyzed on a FACS Canto II Flow Cytometer (BD Biosciences, San Jose, CA, USA) using the FlowJo 10.6 software (Tree Star, Ashland, OR, USA). 

### 3.4. Proliferation Assay

ASCs were plated at a density of 5 × 10^4^ cells/well and cultured in 10% FBS/DMEM for the indicated time. Cell proliferation was assessed using the MTT and trypan blue assays. For the MTT assay, ASCs were plated at a density of 2 × 10^4^ cells/well and cultured in 10% FBS/DMEM for the indicated time. At the indicated time points, MTT was added and incubated for 4 h; the reaction was stopped by replacing the medium with DMSO, and the formazan salts were dissolved. The conversion of yellow thiazolyl blue tetrazolium to purple formazan was measured at 570 nm wavelength. For the trypan blue assay, the cell number was counted using the trypan blue exclusion viable cell assay, trypsinized, resuspended in equal volumes of medium, trypan blue, and counted using a hemocytometer. All assays were performed in triplicate in at least three independent experiments. 

### 3.5. Adipocyte Differentiaton 

For adipogenic differentiation, ASCs were seeded at a density of 2 × 10^5^ cells/well in 6-well tissue culture plates containing 20% FBS/DMEM. The cells were prepared in triplicates. After 24 h, induction was initiated by replacing the culture medium with adipogenic differentiation medium containing 10% FBS, 0.5 mM 3-isobutyl-1-methylxanthine (IBMX), 1 µM dexamethasone, 100 nM insulin, 1 µM rosiglitazone, and 1% penicillin-streptomycin in DMEM. From day 2 until day 6, the culture medium was replaced every 2 days with post-differentiation medium containing 10% FBS, 100 nM insulin, 1 µM rosiglitazone, and 1% penicillin-streptomycin in DMEM. On day 7 of adipogenic differentiation, the differentiated cells were used for Oil Red O staining or RNA isolation.

### 3.6. RNA isolation and Quantitative real time-Polymerase Chain Reaction (qRT-PCR)

Total RNA was isolated from ASCs using TRIzol reagent (Invitrogen), according to the manufacturer’s instructions. The RNA quality was evaluated using a NanoDrop 2000c spectrophotometer (Thermo Fisher Scientific, Waltham, MA, USA). Two micrograms of Total RNA were reverse-transcribed using the High-Capacity cDNA Reverse Transcription Kit (Applied Biosystems, Waltham, MA, USA). qPCR was performed using the Quant Studio 1 Real-Time PCR System (Applied Bio). All experiments were performed in duplicate and *Arbp* was used as an internal control for normalization. The relative gene expression levels were determined using the 2^−ΔΔCT^ method. Primers used in this study are listed in Appendix A. 

### 3.7. Immunoblotting

After adipocyte differentiation, protein was isolated using Pierce RIPA lysis buffer (Thermo Scientific) with 1X protease inhibitor cocktail (BioRad, Hercules, CA, USA). After determination of protein concentration with DC™ Protein Assay (BioRad), proteins were immunoblotted against specific antibodies for adipocyte-specific marker, including PPARγ (Cell Signaling Technologies, #2443, Danvers, MA, USA), C/EBPα (Cell Signaling Technologies, #2295), FABP4 (Cell Signaling Technologies, #3544), and HSP90α/β (Santa Cruz, #sc-13119, Santa Cruz, CA, USA) as control. Primary antibodies were incubated overnight at 4 °C and secondary antibodies were detected using GelDoc (GE Healthcare Amersham Imager 600, Chicago, IL, USA).

### 3.8. Statistical Analysis

All statistical analyses were performed using GraphPad Prism 9 software (GraphPad Software, La Jolla, CA, USA). Data are presented as mean ± SEM. Statistical significance between two mean values was tested using a two-tailed unpaired Student’s *t*-test with the assumption of equal variances. Comparisons among multiple groups were performed using one-way analysis of variance (ANOVA), with Bonferroni post hoc tests to determine *p*-values. Differences were considered statistically significant at *p* < 0.05.

## Figures and Tables

**Figure 1 ijms-23-11839-f001:**
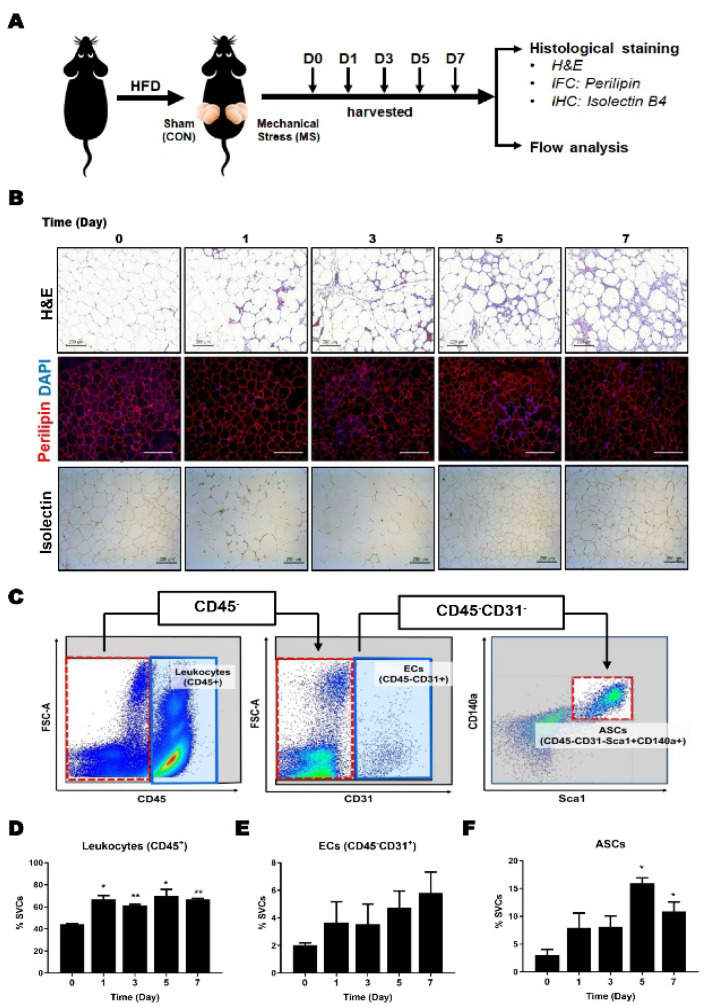
Effects of mechanical stress on adipose tissue histology and cellular composition. Mechanical stresses were performed by needling 25 times with 21-gauge needle on the right unilateral inguinal fat pads of mice. After procedures, fat pads were isolated at different time points (days 0, 1, 3, 5 and 7) and followed by histological or flow cytometry analysis. (**A**) Schematic diagram of experimental design; (**B**) Representative image of H & E (upper), immunostaining of Perilipin (red) and DAPI (blue) (middle) and immunostaining of isolectin B4 (bottom) with 10× magnification; (**C**) Gating strategy for leukocytes, endothelial cells (ECs), and adipose-derived stem cells (ASCs); (**D**) Quantitation of leukocytes (CD45^+^) in adipose tissue; (**E**) Frequency of ECs (CD45^−^CD31^+^) in adipose tissue; (**F**) Frequency of ASCs (CD45^−^CD31^−^CD140a^+^Sca1^+^) in adipose tissues. Data are means ± SEM (*n* = 6). * *p* < 0.05; ** *p* < 0.01 versus day 0.

**Figure 2 ijms-23-11839-f002:**
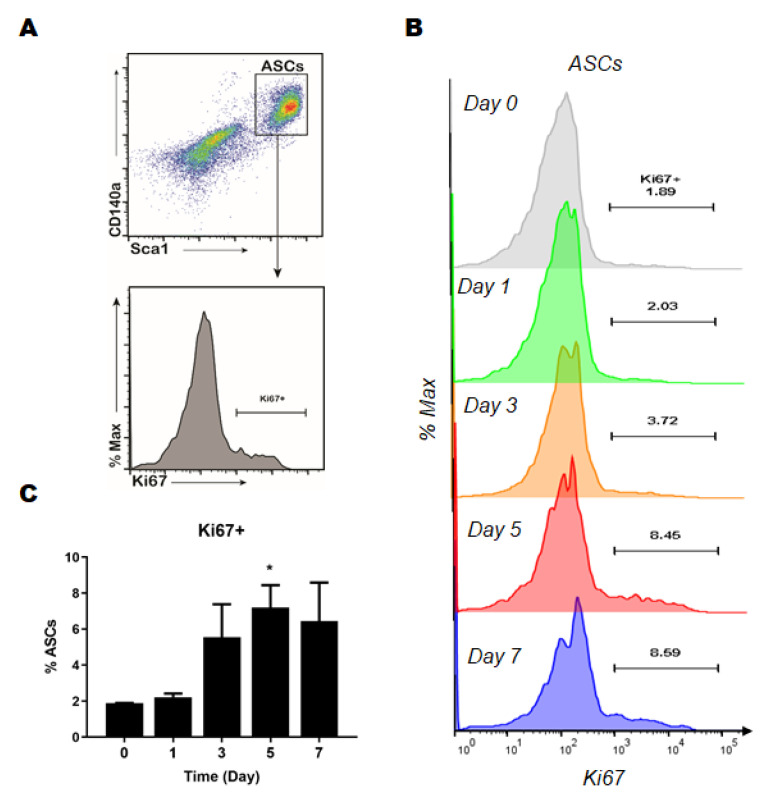
Quantitative analysis of adipocyte-derived stem cells (ASCs) proliferation; (**A**) Gating strategy for flow cytometry analysis; (**B**) Representative flow cytometry profiles showing proliferating (Ki-67^+^) ASCs at day 0 (gray), 1 (green), 3 (orange), 5 (red) and 7 (blue); (**C**) Quantitation of proliferating ASCs (Ki67^+^) in adipose tissue at different time points. Data are means ± SEM (*n* = 4). * *p* < 0.05 versus day 0.

**Figure 3 ijms-23-11839-f003:**
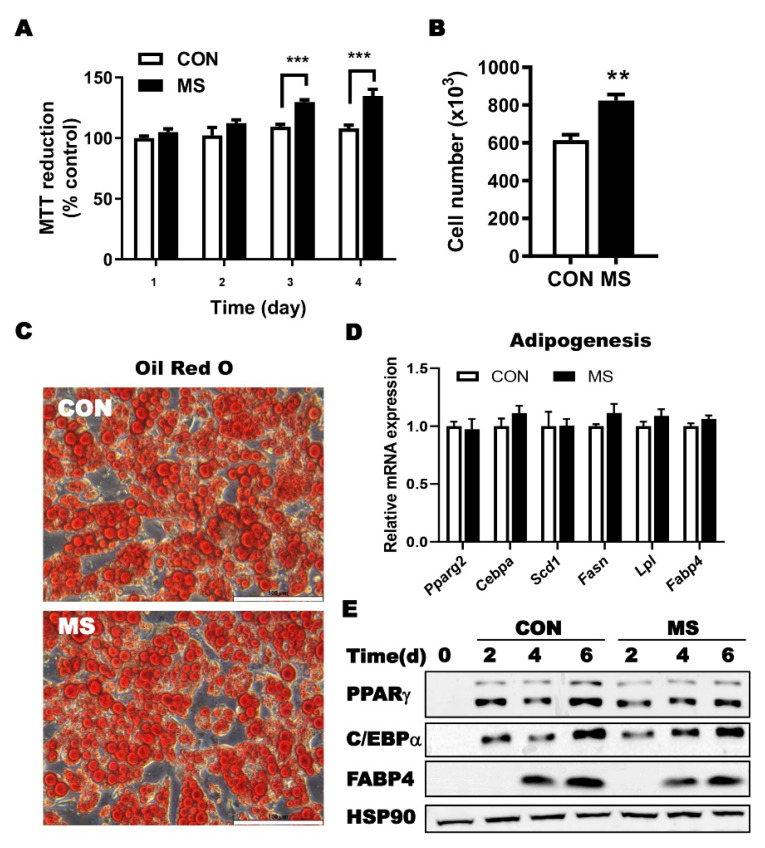
Effects of mechanical stress (MS) on the proliferation and differentiation potential of ASCs. Surgical preconditioning procedures were performed by 25 times needling with 21-gauge needle on the right unilateral inguinal fat pads of mice. SVC were isolated from fat pad at day 5 post-MS or sham procedures and 5 × 10^6^ cells were plated, cultured, and followed by in vitro analysis; (**A**,**B**) MTT reduction and cell number; (**C**) Representative image of Oil Red O staining after adipocyte differentiation of ASCs from MS-treated mice with 40× magnification; (**D**) Gene expression of adipocyte-specific marker genes. (**E**) Immunoblotting of adipocyte-specific protein, including PPARγ, C/EBPα, FABP4, and HSP90 as housekeeping protein. Data are means ± SEM (*n* = 4). ** *p* < 0.01; *** *p* < 0.001 vs. CON.

**Figure 4 ijms-23-11839-f004:**
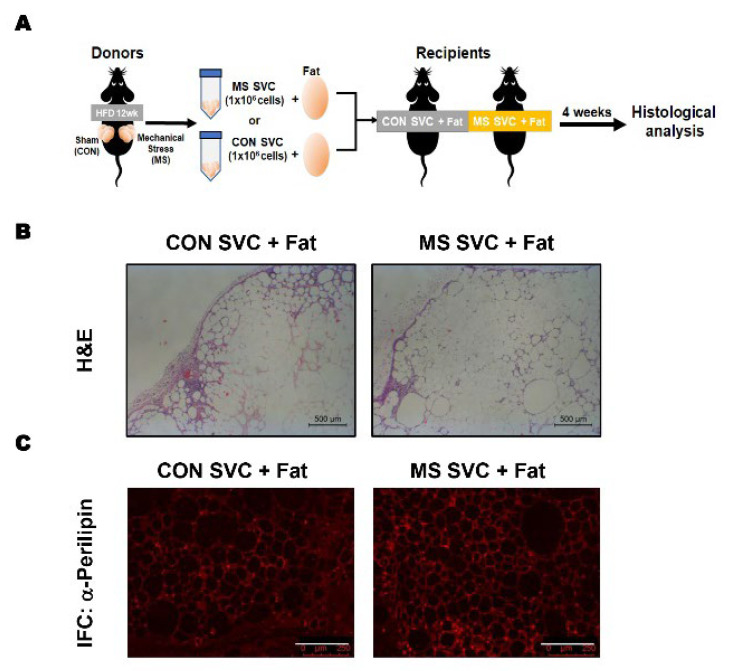
Histological analysis of fat grafts supplemented with SVC from adipose tissue treated with mechanical stress. CAL with SVC obtained from the donor site after surgical preconditioning procedure was performed to examine the efficacy of the obtained SVC (*n* = 6 per group); (**A**) Schematic diagram of experimental design; (**B**) H&E staining (4× magnification); (**C**) Representative immunohistological staining of Perilipin (10× magnification).

## Data Availability

Not applicable.

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
