# Peer review of "Mechanical Stress Improves Fat Graft Survival by Promoting Adipose-Derived Stem Cells Proliferation"

_ijms, 2022, doi:10.3390/ijms231911839_

Round 1

Reviewer 1 Report

In the article: “Mechanical Stress Improves Fat Graft Survival by Promoting Adipose-derived Stem Cells Proliferation” the authors discussed about an innovative technique to obtain an  aspirated fat with enrichment of adipose-derived stem cells (ASCs).

Overall, this manuscript results very interesting, the authors clearly explain the rational of the study and discussed the topic point by point.

However, we would like to invite the authors  to clarify some minor points:

 1.       Please check the check punctuation and spaces;

2.      Into introduction section, the authors describes in general the importance of ASC and their potential application.  The authors should deep the potential applications of stem cells, the possibility to differentiate and the importance to preserve a specific phenotype.  In this respect the following references should be useful:  “Alessio N, Stellavato A, Aprile D, Cimini D, Vassallo V, Di Bernardo G, Galderisi U, Schiraldi C. Timely Supplementation of Hydrogels Containing Sulfated or Unsulfated Chondroitin and Hyaluronic Acid Affects Mesenchymal Stromal Cells Commitment Toward Chondrogenic Differentiation. Front Cell Dev Biol. 2021 Apr 12;9:641529. doi: 10.3389/fcell.2021.641529. PMID: 33912558; PMCID: PMC8072340”;

3.    Among the materials and methods the authors say: “After mechanical stress, the microenvironment of the donor site fat tissue was analyzed in a time-dependent manner on days 0, 1, 3, 5, and 7 (n = 6~8 for each group). Please, describe in which manner they did it;

4.  Figure 1B; the images are not very clear, in particular, for Perilipin immunostaining please indicate in the legend blue and red what mean, and the scale bare is not visible;

5.   Figure 4C; the same of Figure 1B;

6.   Are other specific analyses in order to confirm the cellular phenotype for examples western blotting available? Do you think that gene expression analyses are enough?

Author Response

  1. Please check the check punctuation and spaces;

-> Authors really appreciate the reviewer’s efforts. Based on the suggestion, the authors checked the punctuation and spaces very carefully in the revised manuscript.

  1. Into introduction section, the authors describes in general the importance of ASC and their potential application. The authors should deep the potential applications of stem cells, the possibility to differentiate and the importance to preserve a specific phenotype.  In this respect the following references should be useful: “Alessio N, Stellavato A, Aprile D, Cimini D, Vassallo V, Di Bernardo G, Galderisi U, Schiraldi C. Timely Supplementation of Hydrogels Containing Sulfated or Unsulfated Chondroitin and Hyaluronic Acid Affects Mesenchymal Stromal Cells Commitment Toward Chondrogenic Differentiation. Front Cell Dev Biol. 2021 Apr 12;9:641529. doi: 10.3389/fcell.2021.641529. PMID: 33912558; PMCID: PMC8072340”;

-> Authors really appreciate the reviewer’s constructive criticism. The reviewer’s comments improve the current manuscript better. Based on the comments, the authors added more information about the application of stem cells and the importance of stem cells into the introduction parts of the revised manuscript (L58-64, L72-76).

  1. Among the materials and methods the authors say: “After mechanical stress, the microenvironment of the donor site fat tissue was analyzed in a time-dependent manner on days 0, 1, 3, 5, and 7 (n = 6~8 for each group). Please, describe in which manner they did it;

-> Authors really appreciate the reviewer’s efforts. Authors changed the sentence from original to “~ time-dependent manner on days 0, 1, 3, 5, and 7 by histological staining and flow cytometry analysis (n = 6~8 for each group)” (L319~320 in the revised manuscript).

  1. Figure 1B; the images are not very clear, in particular, for Perilipin immunostaining please indicate in the legend blue and red what mean, and the scale bare is not visible;

-> Authors really appreciate the reviewer’s constructive criticism. The authors provided a better quality picture for figure 1B which contains a scale bar. The authors also edited the figure legend in the revised manuscript.

  1. Figure 4C; the same of Figure 1B;

-> Authors really appreciate the reviewer’s constructive criticism. As mentioned by reviewers, the authors provided a better quality picture for  Figure 4C contains a scale bar. We also edited figure legend in the revised manuscript.

  1. Are other specific analyses in order to confirm the cellular phenotype for examples western blotting available? Do you think that gene expression analyses are enough?

-> Authors really appreciate the reviewer’s constructive criticism. We agree with the reviewer’s opinion about the necessity of other analyses regarding the cellular phenotype and then performed the western blotting experiments of ASCs during the adipocyte differentiation. Consistent with the Oil Red O staining and the mRNA expression (Figure 3C & D), protein abundances of PPARg, CEBPa, and FABP4 during adipocyte differentiation were comparable among groups. These results support that mechanical stimulation did not impair the adipocyte differentiation potential of ASCs. We provided this information in the revised manuscript (Figure 3E and L169-172) as the reviewer’s suggestion.

Reviewer 2 Report

The manuscript entitled “Mechanical Stress Improves Fat Graft Survival by Promoting Adipose-derived Stem Cells Proliferation” described the novel data about the influence of mechanical stress on the potency of adipose-derived stem cells (ASCs). The authors use an interesting approach, modern methods.

The study can be accepted after minor revision.

1)Certainly the use of needles for the face, associated with the appearance of small wounds and possible infection, raises some doubts.

2) The discussion of the research results should be expanded, including by increasing the list of references used, for example: doi.org/10.1080/14686996.2022.2082260, doi.org/10.3390/polym14020344 etc.

Author Response

  • Certainly the use of needles for the face, associated with the appearance of small wounds and possible infection, raises some doubts.

-> Authors really appreciate the reviewer’s efforts to improve the manuscript better. In this study, we applied needling as the mechanical stimulation to the donor fat rather than the recipient site. Since donor fat is usually obtained from the thigh, buttocks, and lower back, the use of a needle for the face where is the recipient site did not happen. Although needling can generate small wounds and possible infection, it has been known as a safe procedure. However, the administration of needling in human patients can cause ethical issues. Thus, finding another mechanical stimulant that can replace needling would be needed for future studies. Clarified the information commented by the reviewer was provided in the discussion section of the manuscript (L290-295).   

2)  The discussion of the research results should be expanded, including by increasing the list of references used, for example: doi.org/10.1080/14686996.2022.2082260, doi.org/10.3390/polym14020344 etc.

 -> Authors really appreciate the reviewer’s constructive criticism and agree with the reviewer’s suggestion. Based on the reviewer’s comment, we added references and provided more information about the approaches to the regulation of stem cell behavior using the biomaterials with nanoparticles and in-deep discussion regarding the underlying mechanisms of stem cell regulation by biophysical approaches in the revised manuscript (L229~237, L249~259, L278~286).